# Venous Thromboembolic Disease in COVID-19, Pathophysiology, Therapy and Prophylaxis

**DOI:** 10.3390/ijms231810372

**Published:** 2022-09-08

**Authors:** Małgorzata Dybowska, Dorota Wyrostkiewicz, Lucyna Opoka, Katarzyna Lewandowska, Małgorzata Sobiecka, Witold Tomkowski, Monika Szturmowicz

**Affiliations:** 1Department of Lung Diseases, National Tuberculosis and Lung Diseases Research Institute, 01-138 Warsaw, Poland; 2Department of Radiology, National Tuberculosis and Lung Diseases Research Institute, 01-138 Warsaw, Poland

**Keywords:** COVID-19, SARS-CoV-2, venous thromboembolism, NETs, immunothrombosis, thromboprophylaxis, anticoagulation treatment

## Abstract

For over two years, the world has been facing the epidemiological and health challenge of the coronavirus disease 2019 (COVID-19) pandemic, caused by the severe acute respiratory syndrome coronavirus 2 (SARS-CoV-2) infection. Growing problems are also complications after the development of COVID-19 in the form of post and long- COVID syndromes, posing a challenge for the medical community, both for clinicians and the scientific world. SARS-CoV-2 infection is associated with an increased risk of cardiovascular complications, especially thromboembolic complications, which are associated with both thrombosis of small and very small vessels due to immunothrombosis, and the development of venous thromboembolism. Low molecular wight heparin (LMHW) are the basic agents used in the prevention and treatment of thromboembolic complications in COVID-19. There is still a great deal of controversy regarding both the prevention and treatment of thromboembolic complications, including the prophylaxis dose or the optimal duration of anticoagulant treatment in patients with an episode of venous thromboembolism.

## 1. Introduction

Coronavirus disease 2019 (COVID-19) is caused by the severe acute respiratory syndrome coronavirus 2 (SARS-CoV-2). So far, 559 million infections have been confirmed around the world, and 6.36 million of those died due to COVID-19 [1].

The clinical spectrum of COVID-19 ranges from asymptomatic or oligo-symptomatic infection, presenting as mild flu-like symptoms, to the life-threatening disease, with acute respiratory distress syndrome (ARDS) and vascular thrombosis [2]. Hyper-activation of the immune system and subsequent cytokine release are responsible for multi-organ damage occurring in the severe course of the disease [2].

Data on the incidence of venous thromboembolic disease (VTE) in patients hospitalized for COVID-19 are inconsistent.

The meta-analysis of 18,000 patients hospitalized for COVID-19 documented overall a pooled incidence of VTE of 17% [3]. Nevertheless, higher VTE incidence, reaching 25–33%, was noted in populations of patients systematically screened for VTE, as well as in those with critical COVID-19, hospitalized in intensive care units (ICU) [3].

In another meta-analysis including 86 studies (33,970 patients) the overall VTE prevalence estimate was 14.1% [4]. In patients without screening the percentage was 9.5%, while it was significantly higher when ultrasound screening was used, reaching 40.3% [4]. Subgroup analysis revealed high heterogeneity, with a VTE prevalence of 22.7% in intensive care unit (ICU) patients and of 7.9% in non-ICU [4]. The prevalence of pulmonary embolism (PE) in non-ICU and ICU patients was 3.5% and 13.7% [4].

The recent meta-analysis of 14 autopsy studies of 749 patients who died of COVID-19 documented acute pulmonary embolism in 30% of them [5].

Thus, the possibility of VTE should be taken into account in every COVID-19 patient, and especially in those requiring hospitalization due to a severe or critical course of the disease.

The difficulties in the recognition of VTE in the course of COVID-19 are caused by the common symptomatology of COVID-19 interstitial pneumonia and pulmonary embolism (PE), as well as the frequent coincidence of pulmonary and vascular disease.

The optimal treatment of COVID-related VTE, as well as the most effective prophylactic regimens, are still under debate.

In the present review, the current status of knowledge concerning the pathophysiology of COVID-19 induced VTE, as well as its recognition, treatment and prophylaxis, is presented.

## 2. Clinical Vignette

43-years old, obese male (104 kg, BMI 31 kg/m^2^), with a history of bronchial asthma, was admitted to a private hospital due to SARS-CoV-2 infection, self-confirmed by an antigen test. He denied previous anti-COVID-19 vaccination. The patient reported fever up to 40 °C for 10 days, dyspnoea, weakness, a dry cough, headache, nausea and diarrhea. He used antipyretic drugs (paracetamol, ibuprofen) and azithromycin for 3 days—without any improvement to his health. Three days before hospitalization, small hemoptysis and increasing breathlessness developed.

On admission, hemoglobin oxygen saturation was 80%. High-resolution computed tomography (HRCT) revealed interstitial pneumonia, characteristic of COVID-19, covering approximately 50–70% of lung parenchyma. C-reactive protein (CRP) was 361 mg/L; (N < 5 mg/dL). The patient received oxygen through a nasal cannula with a flow of 5 L/ min, empirical intravenous antibiotic therapy with amoxycycline plus clavulanic acid and levofloxacin, low molecular weight heparin (LMWH) in prophylactic dose.

Because of significantly elevated D-dimers (4528 ng/mL; N < 500 ng/mL), an urgent chest computed tomography pulmonary angiography (CTPA) was performed, which showed right-sided thrombus localized in the lower lobe artery and in the segmental arteries 6, 8, 9, 10.

LMWH dose was increased to therapeutic range (1 mg/kg sc, twice daily).

At that time, the PCR test for SARS-CoV-2 (ADVANCED ONE STEP FAST CoVi19 KIT TWO GENE SET, GeneMe Sp.z.o.o.), performed twice with an interval of 24 h, was negative.

The patient was admitted to the Ist Department of Lung Diseases, National Tuberculosis and Lung Diseases Research Institute, for further treatment.

On admission he reported significant weakness, a dry, tiring cough and marked exercise dyspnoea. His body temperature was 36.8 °C, blood pressure (BP) 116/68 mmHg, heart rate (HR) 100/min, oxygen saturation 87% (measured during oxygen therapy, 5 L/min, via nasal cannula). On auscultation, numerous crackles were heard over both lungs.

The laboratory tests results are shown in Table 1. A marked increase in plasma D-dimer and serum ferritin concentrations were found. 

A chest X-ray revealed a reduced lung volume, and both-sided parenchymal lesions, localized in the middle and lower lung fields, predominating in the left lung (Figure 1a).

Echocardiography showed features suggestive of the intermediate probability of pulmonary hypertension: the tricuspid valve regurgitation peak gradient (TVPG) was 32 mmHg, the pulmonary artery acceleration time (AcT)-80 ms, without any signs of right ventricular overload. An ultrasound examination of the deep veins of the lower limbs excluded thrombosis.

Due to respiratory failure, oxygen supplementation was increased to 7 L/min, salbutamol and budesonide in nebulizations were used, and dexamethasone 8 mg iv once daily was started. Antibiotics and anticoagulant therapy were continued. 

As the diagnosis of SARS-CoV-2 was not confirmed by PCR, antibodies against SARS-CoV-2 were determined using the indirect chemiluminescence method (ALINITY I, Abbott 2). The result was positive in the IgM class 44.4 S/C (negative < 1.0) and in the IgG class 6.0 S/C (negative < 1.4), which indicated the early stage of COVID-19 disease.

During further hospitalization, a gradual improvement in the patient’s health was observed. The oxygen flow was reduced proportionally to the improvement of blood oxygenation, until its complete discontinuation after two weeks of treatment.

The full dose of enoxaparin was reduced to 1 mg/kg sc once daily. After clinical improvement, dexamethasone was switched to oral methylprednisolone (Metypred) 16 mg/day, and gradually reduced. After three months, LMWH was stopped and rivaroxaban was administered at a dose of 20 mg/day.

At a follow-up examination, after six months, the patient was in good clinical status. The physical examination revealed single crackles over the lower lung fields. Resting blood gases results, as well as CRP, D-dimer and ferritin concentrations had normalized. (Table 1).

The control chest X-ray showed an almost complete regression of parenchymal lung disease, compared to the initial examination (Figure 1b). A follow-up CTPA showed complete regression of pulmonary embolism (Figure 2a,b); moreover, significant regression of parenchymal lung disease was found (Figure 3a,b).

Steroids were stopped after six months of treatment. Rivaroxaban was reduced to 10 mg/day for the next six months, and then the anticoagulant treatment was discontinued. 

## 3. SARS-CoV-2 Infection and the Immune Response to Pathogen

SARS-CoV-2 is a single-stranded RNA virus, made up of the nucleocapsid, membrane, envelope and spike (S) protein. S protein is assembled as a homotrimer and is inserted in multiple copies into the membrane of the virion, giving it its crown-like appearance [6]. The S protein consists of two non-covalently associated subunits: S1 and S2 [6].

SARS-Cov-2 binds with S1 to angiotensin, converting enzyme type-2 (ACE-2) receptor of human cells and the S2 subunit anchors the S protein to the cell membrane [6]. ACE-2 receptors are broadly expressed on the surface of airway epithelial cells. Moreover, they are widely distributed in vascular systems of the heart, the respiratory system, the pancreas, the liver and the placenta, which makes these organs targets for the SARS-Cov-2 infection [6]. ACE-2 receptors are presented on endothelial cells of both arterial and venous vessels, as well as in the microcirculation [6].

Vasculitis and endothelial cell damage may develop from direct viral invasion, or indirectly, in the course of hypoxemia and the inflammatory response of the host immune system. The endothelial cell damage results in the influx of the morphological blood components as well as inflammatory cells and mediators to the exposed subendothelium.

Platelets are first recruited and, once activated, they express adhesion receptors, including P-selectin, platelet factor 4 and release chemokines that mediate the recruitment of innate immune cells including neutrophil-activating peptide-2, which stimulate further recruitment and activation of neutrophils and monocytes/macrophages [7,8].

Neutrophils and macrophages are the first-line defense cells eliminating microorganisms through unspecific mechanisms, such as oxidative stress, and the production of various antimicrobial factors. The role of neutrophils in controlling infection include the phagocytosis of pathogens, the proteolytic activity of substances included in azurophilic granules, and the formation of neutrophil extracellular traps (NETs). NETs are large, extracellular, cobweb-like structures formed from DNA fibers and histones with proteolytic substances from degranulated azurophilic bodies, such as neutrophilic elastase, myeloperoxidase, cathepsin G, lactoferrin and others. They are very effective in catching and destroying pathogens, thus locating the infection and limiting its spread [9].

Monocytes that reach the extravascular space become activated and differentiate into mature tissue macrophages. In addition, the degranulation products released by neutrophils also have a significant influence on the transformation of monocytes into macrophages.

Macrophages can exhibit pro- or anti-inflammatory properties, and this is determined by stimuli from their local microenvironment [10].

Based on their cytokine/mediator profile, two distinct types of macrophages are designated: M1 macrophages, differentiated under the influence of granulocyte-macrophage colony-stimulating factor (GM-CSF), with a pro-inflammatory signature, and M2 macrophages, differentiated under the influence of macrophage colony-stimulating factor (M-CSF), with an anti-inflammatory signature [10].

Pro-inflammatory cytokines secreted by M1 macrophages include IL-1β, IL-6, IL-12, IL-18, IL-23, IL-27, and tumor necrosis factor (TNF) [10].

Anti-inflammatory properties of M2 macrophages are determined by the production of cytokines, i.e., IL-4, IL-10, IL-13, IL-19, and growth factors, i.e., transforming growth factor (TGF)-β and vascular endothelial growth factor (VEGF) [10].

At the inflammatory phase of the SARS-CoV-2 infection, proinflammatory cytokines, such as IL-6, interferon (IFN)-γ and IL-2, dominate [11].

In severe COVID-19, the host immune system can be involved in the development of a lethal inflammatory response, known as cytokine release syndrome, or cytokine storm [12].

The underlying mechanisms responsible for the unrestricted release of inflammatory agents are still unclear, but several hypotheses exist.

One of them concerns pyroptosis, a highly inflammatory form of lytic-programmed cell death (apoptosis), stimulated by viral replications. In COVID-19 patients, pyroptosis triggers the release of pro-inflammatory cytokines and affects macrophage and lymphocyte functions, causing peripheral lymphopenia [12].

Another hypothesis is associated with adaptive immunity and the production of neutralizing antibodies against the surface antigen of the virus [12].

Various NETs components are also involved in the pathogenesis of the cytokine storm in SARS-CoV-2 infection [13].

## 4. Pathogenesis of Thrombotic Complications in COVID-19 Patients

Healthy endothelium has anticoagulant and antithrombotic properties. This is due to the regulated secretion of antiplatelet agents, including prostacyclin and nitric oxide. In addition, the surface of the endothelium is the site for inactivation of thrombin by antithrombin, and its conversion to a coagulation inhibitor by interaction with thrombomodulin. Endothelial cells are also the source of circulating the tissue-type plasminogen activator (TPA) and its inhibitor, and the tissue factor (TF) pathway inhibitor.

Endothelial cell injury following SARS-CoV-2 infection results in the disruption of physiological anticoagulant function and the development of a more procoagulant and pro-thrombotic phenotype [14].

The enhancement of coagulation processes in COVID-19 may be a consequence of increased clotting mechanisms, decreased fibrinolysis and the immune reactions that trigger the immunothrombosis.

The term immunothrombosis, originally described by Engelmann and Massberg, referred to an intrinsic effector pathway of innate immunity triggered by pathogens and injured cells to reduce the spread and survival of the invading pathogen [7].

From that point of view, immunothrombosis may be considered to be a beneficial mechanism of intravascular immunity, but when immunothrombosis is uncontrolled, it causes the dysregulated activation of the coagulation cascade, leading to microthrombosis and/or disseminated intravascular coagulation [7].

Immunothrombosis has been proposed to be an important pathological mechanism in patients with COVID-19, whereby innate immune cell activation, excessive coagulation and endothelial dysfunction contribute to the observed prothrombotic state [11].

The basic feature of immunothrombosis depends on the interaction between the hemostatic and immune systems, in particular monocytes, macrophages and neutrophils.

The subsequent release of cytokines contributes to the hypercoagulable state. Pro-inflammatory cytokines, in particular IL-6, induce platelet activation and aggregation. Cathepsin G, a serine protease produced by neutrophils, also activates platelets [11]. Important players in dysregulated immunothrombosis in COVID-19 are NETs [15].

The release of NETs initiates the thrombotic processes in small vessels, both veins and arteries, which in turn may lead to the damage of many organs such as the lungs, heart, kidneys and others [16].

NETosis enhances the activity of the coagulation system by increasing fibrin deposition [17].

Histones present in NETs, especially histones H3 and H4, can enhance thrombin generation by reducing thrombomodulin-mediated protein C activation and by directly activating platelets [17]. Moreover, NETs-induced platelet aggregation is observed, resulting in thrombocytopenia observed in patients with severe infections [13].

In this way, NETs have been shown to play a significant role in the pathogenesis of thrombosis by activating platelets, recruiting neutrophils to the endothelial wall, and activating the intrinsic and extrinsic coagulation cascade [13].

As previously mentioned, SARS-CoV-2 destroys the endothelium and leads to cell apoptosis, which causes the over-expression of clotting factors VII, VIII and TF, the release of von Willebrand Factor activating intrinsic and extrinsic coagulation pathways.

The hypercoagulable state is also due to fibrinolytic abnormalities such as increased expression of plasminogen activator inhibitor-1 (PAI-1) or downregulation of anti-thrombin [18].

In addition, neutrophil elastase released from neutrophils can proteolise anticoagulant factors i.e., the antithrombin and tissue factor pathway inhibitor [19].

The impact of SARS-CoV-2 infection on endothelial cells and platelets is shown in Figure 4a.

The influence of the SARS-CoV-2 infection on coagulation processes is shown in Figure 4b.

The cumulative response of the immune system to SARS-CoV-2, both through inflammation and stimulation of prothrombotic proteins, is likely to be a major contributor to hypercoagulability in COVID-19 [11].

It is likely that all the mentioned mechanisms, through their synergistic action, lead to the development of COVID-19-dependent thrombophilia and can cause thrombosis of both small and large vessels.

## 5. Laboratory Tests in Patients with COVID-19

Inflammatory hyper-responsiveness in SARS-CoV-2 infection results in the increase of C-reactive protein (CRP), interelukin-6 (IL-6) and fibrinogen concentration [20]. Profound systemic inflammation, referred to as SIRS (Systemic Inflammatory Response Syndrome), is characterized by an increase in ferritin, LDH (lactate dehydrogenase) and fibrin degradation product (D-dimer) concentration [20].

In the patients who progress to SIRS and develop profound hypoxemia, high levels of pro- inflammatory cytokines (IL-6, IL-1β, IL-18 and granulocyte-macrophage colony-stimulating factor) are found [17]. This has been referred to as a cytokine storm [17].

Increased prothrombotic readiness in many COVID-19 patients may be indicated by a high concentration of D-dimer, an increased concentration of fibrinogen and/or a low concentration of antithrombin [21].

In addition, laboratory tests show a prolongation of prothrombin time (PT) and activated partial thromboplastin time (APTT) and an increased concentration of factor VIII [11,21].

Among routine tests, the best marker to assess the risk of thrombotic complications in a patient with COVID-19 is the determination of D-dimer [21,22,23].

In a retrospective cohort study, in patients with confirmed COVID-19 hospitalized in two French centers, it was stated that the negative predictive value of a baseline D-dimer level < 1.0 µg/mL was 90% for VTE and 98% for pulmonary embolism (PE). The positive predictive value for VTE was 44% and 67% for D-dimer level ≥ 1.0 µg/mL and ≥3 µg/mL, respectively [24].

In the meta-analysis of studies including hospitalized COVID-19 patients, those developing VTE had higher D-dimer levels (weighted mean difference, 3.26 µg/mL; 95% CI, 2.76–3.77) than non-VTE patients [4].

On the other hand, an increased D-dimer level can be observed in the absence of VTE, as part of COVID-19-related coagulopathy [25].

The appearance of a dynamic decrease in hemoglobin oxygen saturation and an increase in the concentration of D-dimer are often indicators of the development of small pulmonary vascular thrombosis in COVID-19 patients.

In addition, a high concentration of D-dimer is one of the factors that worsen the prognosis in COVID-19 [22,23].

## 6. Micro-Thrombosis and Venous Thromboembolism in COVID-19 Patients

Pulmonary micro-thrombosis is one of the basic mechanisms involved in the pathogenesis of the critical COVID-19 [26]. Thrombosis of small and very small pulmonary vessels most likely enhances the inflammatory process and destruction of the endothelium, and consequently intensifies diffuse alveolar damage (DAD) [26].

These thrombotic changes occurring in situ in small and very small pulmonary vessels are not available in imaging studies of the pulmonary vasculature, such as CTPA or angiography of the pulmonary arteries.

Apart from thrombotic changes in pulmonary vessels in situ, patients infected with SARS-CoV-2 also develop VTE more frequently. All elements of the Virchow triad are involved in its pathogenesis: endothelial damage, coagulation disorders and slowing blood flow, due to immobilization during severe course of the disease. 

It is worth emphasizing that the common risk factors for both severe course of COVID-19 and VTE are older age, obesity, previous heart or respiratory failure.

## 7. Prophylaxis of Thrombotic Complications

LMWH are the basic agents used in the prevention and treatment of thromboembolic complications in COVID-19 disease.

The beneficial influence of LMWH results both from their anticoagulant properties, and their anti-inflammatory activity [27,28,29].

LMWH exhibit anti-inflammatory properties by neutralizing chemokines and cytokines, inhibiting leukocyte migration, neutralizing NETs and inhibiting the activity of heparanase, responsible for vascular permeability [27,28,29]. Moreover, they have antiviral activity by influencing the binding of the viral S protein to the ACE-2 receptor, which may be of particular importance at the initial stage of the disease [27,28,29]. Therefore, LMWH are currently considered the drugs of choice, both in the period of active infection and in cases with persisting chronic inflammation [27,28,29].

In the available literature regarding the use of thromboprophylaxis in patients with COVID-19 there are many discrepancies. So far, there is no consensus on the recommended drugs and their doses. A common practice in patients hospitalized for COVID-19 is to use a prophylactic dose of LMWH. In order to optimize the dosage of LMWH used in thromboprophylaxis, the following factors should be taken into account: renal function, body weight, platelet count, fibrinogen concentration, and anti-Xa results [30,31].

In individual cases, especially in patients with a particularly high risk of thrombotic complications, increasing the dose of LMWHs in thromboprophylaxis may be considered [30,31,32]. The routine use of therapeutic doses of LMWH in this indication is not recommended.

However, this issue still remains open to discussion.

A recently published meta-analysis of seven studies summarizing the efficacy and safety of standard versus intermediate LMWH doses for thromboprophylaxis concluded that the use of intermediate doses appeared to be safe and was associated with additional survival benefits [33]. The limitation of the study was the fact that most of the data came from retrospective analyzes.

Several randomized controlled trials have evaluated the role of therapeutic doses of heparin in reducing VTE events or mortality in patients hospitalized for COVID-19 [30].

Randomized clinical trials comparing the efficacy and safety of a prophylactic dose of LMWH versus the therapeutic dose in hospitalized patients with severe or moderate COVID-19 (REMAP-CAP, ATTACC, and ACTIV-4A) demonstrated the benefit of therapeutic doses for patients in better condition (not critically ill, without the need for organ support), especially in the group with a high concentration of D-dimer (≥2 times the upper limit of the normal range [ULN]) [34].

Patients with high levels of D-dimer were generally older and had more comorbidities [33]. Major bleeding occurred in 1.9% of patients receiving the therapeutic dose of LMWH and in 0.9% of patients receiving standard thromboprophylaxis [34].

In the RAPID trial, which enrolled patients with elevated D-dimer levels and hypoxemia, a therapeutic dose of heparin reduced all-cause mortality [35].

The HEP-COVID trial enrolled patients who required supplementary oxygen and had a D-dimer value > 4 times ULN or a sepsis-induced coagulopathy score of ≥4. Therapeutic-dose LMWH reduced major thromboembolism and death risk, compared with the institutional standard heparin thromboprophylaxis. The treatment effect was not seen in ICU patients [36]. 

According to the latest update of the National Institutes of Health (NIH) and American Society of Hematology (ASH) guidelines, a therapeutic dose of heparin is recommended for patients with D-dimer level above ULN who require low flow oxygen therapy and do not have an increased risk of bleeding [30,31].

The latest International Society on Thrombosis and Haemostasis (ISTH) guidelines, issued in July 2022, gave a strong recommendation for non-critically ill patients hospitalized for COVID-19: (a) for the use of prophylactic dose of low-molecular weight heparin or unfractionated heparin (LMWH/UFH) (b) for the use of therapeutic dose LMWH/UFH in preference to prophylactic dose, in selected patients belonging to this group, but (c) against the addition of an antiplatelet agent [32].

Based on clinical trial exclusion criteria, contraindications for the use of therapeutic anticoagulation in patients with COVID-19 are: a platelet count <50 × 10⁹/L, hemoglobin <8 g/dL, the need for dual antiplatelet therapy, bleeding within the past 30 days that required an emergency department visit or hospitalization, history of a bleeding disorder, an inherited or active, acquired bleeding disorder [30].

The benefits of using therapeutic anticoagulation with heparin have not been found in critically ill patients requiring ICU treatment [37]. Probably the therapeutic doses of heparins are not able to influence the cascade of inflammation, thrombosis and organ damage in patients with end-stage inflammatory disease, however, such therapy may be considered in patients with moderate disease course [30,31].

There is no sufficient evidence to recommend thromboprophylaxis for SARS-CoV-2 infected patients during isolation at home [30,31].

ASH guidelines advise not starting anticoagulation for acutely ill COVID-19 outpatients [38].

The ISTH panel gave a weak recommendation regarding the use of sulodexide in non-hospitalized patients. It is based on one study, single-center placebo-controlled, which showed that use of sulodexide may reduce the risk of hospitalization and possibly also the need for oxygen supplementation. In this study, 243 non-hospitalized patients with COVID-19 were randomized to oral sulodexide 500 lipase-releasing units twice daily or placebo. The inclusion criterion was a high risk of COVID-19 progression, as defined by the COVID-19 Health Complications (C19HC) calculator, which takes into account age, body mass index, smoking and chronic comorbidities. The trial showed a statistically significant decrease in the risk of hospitalization with an absolute risk reduction (ARR) of 11.7%, a borderline significant reduction in oxygen supplementation, a non-significant decrease in all causes of mortality, and no indication of harm associated with the treatment. The trial did not demonstrate decreased risk of thrombotic events. Overall, the trial supports the effectiveness and safety of sulodexide in outpatients with COVID-19 [32,39]. These results need to be confirmed in future studies [32].

Patients diagnosed with mild-to-moderate COVID-19 not requiring hospitalization do not benefit from starting anticoagulation treatment, either as thromboprophylaxis or to prevent progression of COVID-19. This recommendation was based on a very low event rate in stable outpatients, observed in ACTIV-4b RCT, which compared placebo, aspirin, or two different doses of apixaban in ambulatory patients older than 40 years [30,31].

The use of direct oral anticoagulants (DOAC) in the prevention and treatment of thromboembolic complications in SARS-CoV-2 infected patients is controversial, due to their possible interactions, especially with antiviral drugs used in COVID-19 therapy.

Rivaroxaban and apixaban have a significant drug interaction with ritonavir. Co-administration will increase the concentration of apixaban or rivaroxaban and may increase the risk of bleeding [31]. Dexamethasone is an inducer of CYP3A4 and the extent of the drug interaction with direct oral anticoagulants is unknown [31].

As shown in the ACTION study comparing the use of a 20 mg dose of rivaroxaban in patients hospitalized for COVID-19 with elevated D-dimer levels, then continued for 30 days after hospital discharge, versus a prophylactic dose of enoxaparin used during hospitalization, extended rivaroxaban use did not improve the effectiveness of therapy. Moreover, the increase of bleeding complications was observed in the group treated with rivaroxaban compared to those treated with the prophylactic dose of LMWH [40].

The risk of VTE following hospital discharge appears to be low and similar to risks following hospital discharge for other medical conditions, based on observational studies [31]. Nevertheless, in Long COVID-19, vascular thrombosis was documented 30 days after the disease onset in up to 2.5% of patients, and VTE- in 0.6% [41].

The MICHELLE trial showed a significant reduction (relative risk 0.33; 95% CI, 0.13–0.90) in post-discharge thrombotic events (including screen-detected DVT and PE at day 35) and death after 35 days of treatment with rivaroxaban 10 mg per day, compared to no anticoagulation in a population of patients selected for increased VTE risk based on the IMPROVE/IMPROVE-DD score (randomized 320 patients with COVID-19 who had an IMPROVE score of ≥4 or a score of 2 to 3 with a D-dimer > 500 ng/mL) [42].

However, both ASH and NIH guidelines do not recommend the routine use of post-discharge anticoagulation, recognizing that there is insufficient evidence to recommend either continuing or discontinuing anticoagulation after hospital discharge, unless another indication for VTE prophylaxis exists. The decision to continue post-discharge VTE prophylaxis for patients with COVID-19 should include consideration of an individual patient’s risk factors for VTE, and bleeding risks [30,31].

ISTH guidelines suggest a weak recommendation prophylactic dose of rivaroxaban for selected patients after discharge [32].

More clinical trials are needed to answer this question.

Table 2 summarizes the recommendations for thromboprophylaxis in patients with COVID-19 without evidence of VTE.

It is emphasized that all patients with COVID-19 should be educated on the signs and symptoms of VTE at hospital discharge and advised to seek urgent medical attention should these develop [31].

It should be noted that from a theoretical point of view, antiplatelet therapy should have a beneficial effect in patients with severe COVID-19 through several mechanisms, including inhibition of platelet aggregation, reduction of platelet-derived inflammation, and blocking of thrombogenic NET’s [43]. However, in the observational studies performed so far, and in the large, randomized trial (RECOVERY) involving over 14,000 patients, aspirin at a dose of 150 mg/day was not associated with a reduction in mortality in patients not requiring mechanical ventilation [44].

In the group treated with aspirin, fewer thromboembolic complications were observed, but the frequency of major bleeding was significantly higher [44].

Another antiplatelet agent, dipyridamole, may have a therapeutic role. Apart from its antiplatelet function, it suppresses inflammation, and exhibits antiviral activity prevent NETosis by promoting 3′,5′-cyclic adenosine monophosphate (cAMP) generation in neutrophils. A trial of dipyridamole (NCT04391179) has been completed, but the results have not been published yet [17].

The ability to attenuate NET formation is also exhibited by another antiplatelet drug, ticagrelol. Accordingly, two trials of ticagrelor are ongoing (NCT02735707, NCT04518735) [17].

## 8. Treatment of VTE Patients

In cases with confirmed venous thromboembolism (acute pulmonary embolism and/or deep vein thrombosis), the current treatment guidelines should be followed. However, the preferred treatment schedule should be a therapeutic dose of LMWH. It is worth emphasizing that LMWH have no significant interactions with drugs used in COVID-19 treatment.

Due to epidemic reasons, the use of unfractionated heparin should be limited in COVID-19 patients because of the need to monitor anticoagulation, which exposes staff to more frequent contact with infected patients [30,31].

Use of DOAC needs caution because of its interactions with some specific antiviral drugs (for example, lopinavir and ritonavir), which increase the risk of bleeding [30,31]. Therefore, the use of DOAC in hospitalized patients is not recommended. Treatment with DOAC may be considered after ICU discharge [30,31].

So far, there are no data on therapeutic dose adjustments in cases of acute renal failure requiring continuous renal replacement therapy, severe thrombocytopenia, or extreme (low or very high) body weight, which are common in hospitalized patients, especially in ICU.

There is still disagreement on the duration of anticoagulation therapy for a VTE episode in the course of a SARS-CoV-2 infection. The therapy should be continued for at least 3 months, nevertheless some authors advise extending treatment to 6-12 months. 

This and many other issues are still not answered, such as:-What parameters should be assessed before deciding to discontinue anticoagulant therapy?-Should patients with a VTE event related to COVID-19 infection be tested for congenital or acquired thrombophilia?-Should patients with a VTE incident be screened for recurrence that would be a revelator for thrombotic complications?

So far, we do not know the answer to these and many other questions. 

These problems require further randomized trials.

## 9. Summary

Understanding the pathophysiological mechanisms of COVID-19 has been crucial for better diagnostic and treatment of this disease. Severe COVID-19 is a multisystem syndrome in which the vascular endothelium is the most damaged organ. It’s known that SARS-CoV-2 infection is associated with an increased risk of cardiovascular complications, especially thromboembolic complications. The underlying cause of these disorders is the phenomenon of immunothrombosis. Anticoagulant therapy and immunomodulatory agents are likely to be necessary to reduce the hyperinflammatory and prothrombotic conditions. LMWH are the basic form of prophylaxis of thromboembolic complications in patients with COVID-19. Evidence from clinical trials is still limited and recommendations for anticoagulation therapy are based on an individual risk profile.

## 10. Conclusions

Thromboembolic complications are a common problem in patients with COVID-19. More randomized controlled trials on the prevention and treatment of thrombotic disorders in patients with COVID-19 are needed.

## Figures and Tables

**Figure 1 ijms-23-10372-f001:**
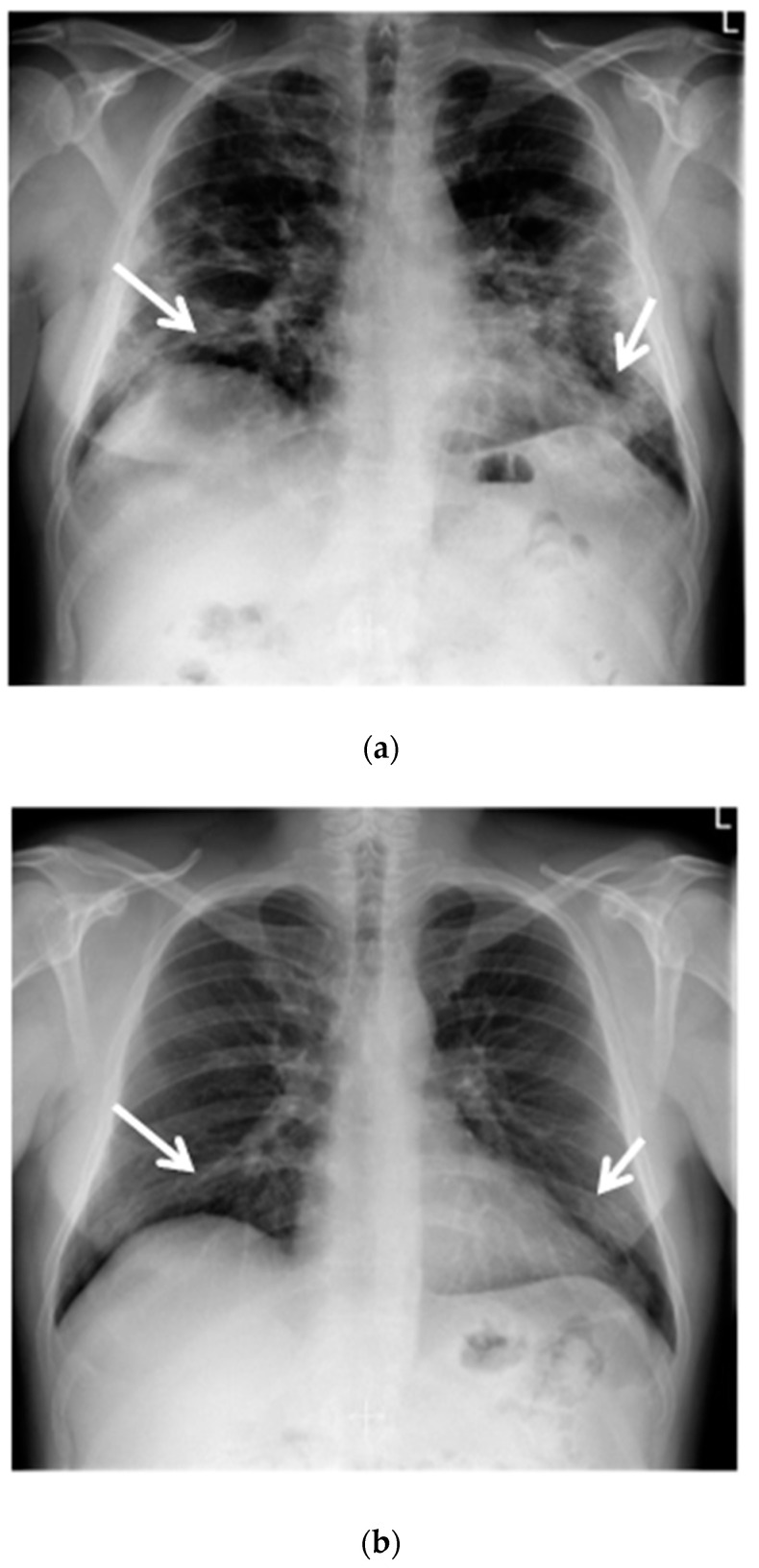
(**a**) Chest X-ray shows opacities due to COVID-19 dominating peripherally in the lower zones of the lungs. Reduction in the volume of the lungs (white arrows); (**b**) Chest X-ray after six months of follow-up shows an almost complete regression of lung opacities (white arrows).

**Figure 2 ijms-23-10372-f002:**
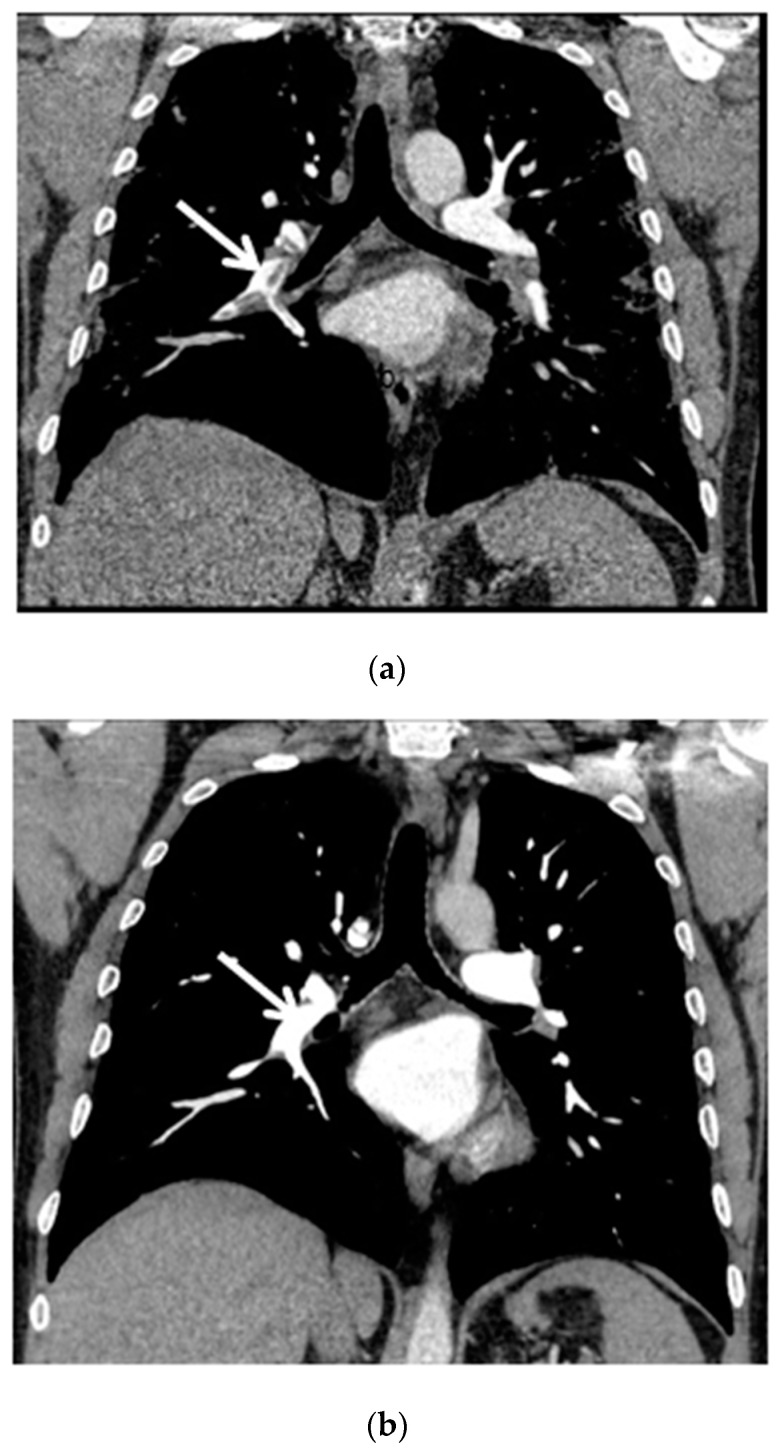
(**a**) Chest CT scan (mediastinal window, coronal view) shows pulmonary embolism that affects the right pulmonary artery, lobar arteries of the right lower and upper lobes and interlobar pulmonary artery (white arrow); (**b**) Chest CT scan after six months of follow-up shows complete resolution of thrombosis (white arrow).

**Figure 3 ijms-23-10372-f003:**
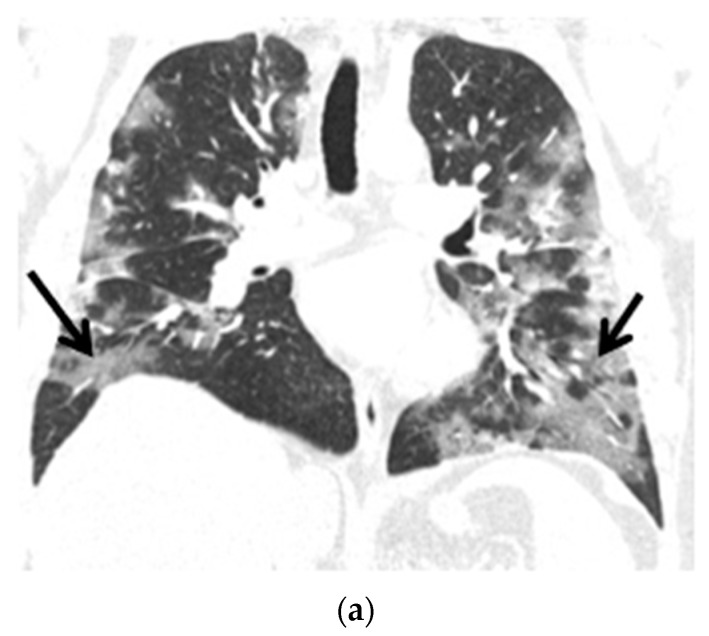
(**a**) Chest CT scans (lung window, coronal view) show patchy ground-glass opacities in accordance with COVID-19 dominant in the peripheral zones of the lower lungs (black arrows); (**b**) Chest CT scans (lung window, coronal view) after six months of follow-up show resolution of lung lesions (black arrows).

**Figure 4 ijms-23-10372-f004:**
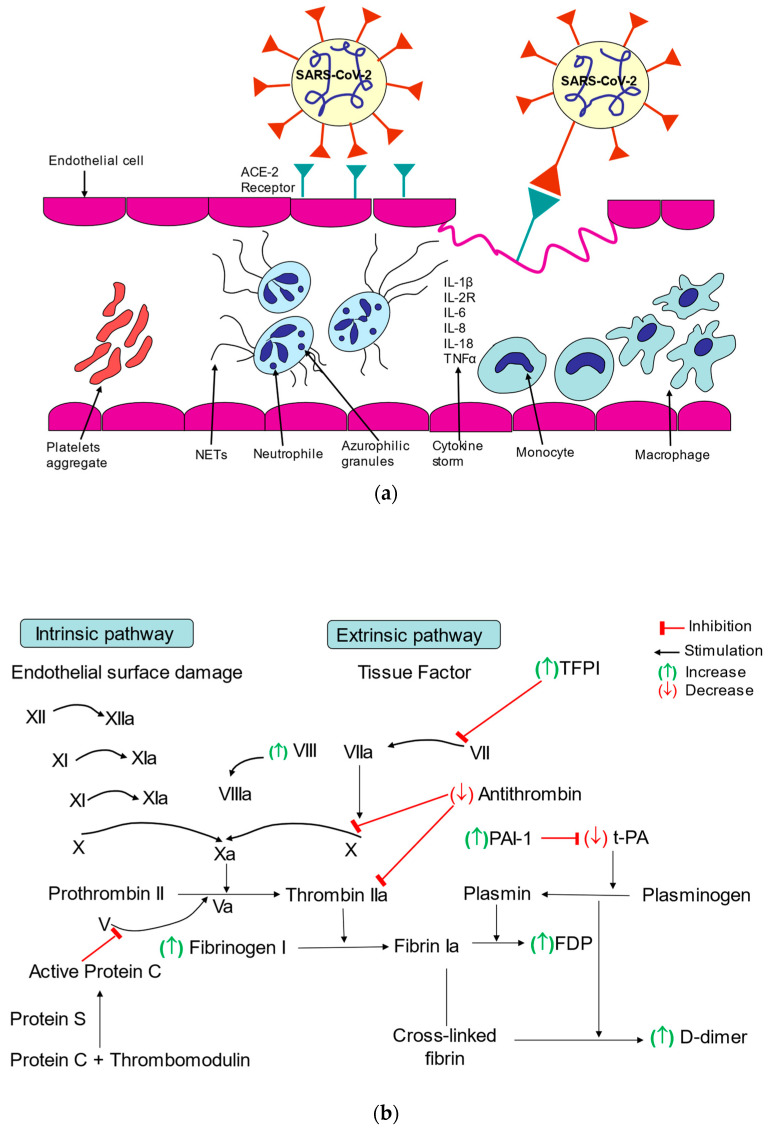
(**a**) The impact of SARS-CoV-2 infection on endothelial cells and platelets; (**b**) Influence of SARS-CoV-2 infection on coagulation processes.

**Table 1 ijms-23-10372-t001:** Laboratory tests results.

	On Admission to Pulmonary Department	At 6 Months Follow-Up	Reference Range
WBC (×10^9^/L)	10.2	6.11	3.98–10.04
CRP (mg/L)	55.6	2.9	<5
D-dimer (ng/mL)	14,298	166	<500
NT-proBNP (pg/mL)	334.1	157	0–125
Ferritin (ng /mL)	753.38	121.6	21.81–274.66
Capillary blood gases	(oxygen 5 L/min)	(no oxygen therapy)	
PaO_2_ (mmHg)	52.5	71.2	65–90
PaCO_2_ (mmHg)	30.8	40.6	35–45
SaO_2_ (%)	90.4	94.7	92–98

WBC—white blood cell count, CRP—C-reactive protein, NT-proBNP—N-terminal-pro-brain natriuretic peptide, PaO_2_—partial oxygen pressure, PaCO_2_—partial carbon dioxide pressure, SaO_2_ hemoglobin oxygen saturation.

**Table 2 ijms-23-10372-t002:** Summary of recommendations for thromboprophylaxis in patients with COVID-19 without evidence of VTE.

Guidelines	ASH	NIH	ISTH
Last update [references]	January 2022 [31]	May 2022 [30]	July 2022 [32]
Non-hospitalized patients	Against	Against	Sulodexide 500LU twice daily for patients at higher risk of disease progression
Hospitalized, non-ICU patients, require low-flow oxygen	LMWH or UFH in therapeutic dose	LMWH or UFH in therapeutic dose	LMWH or UFH in therapeutic dose
ICU patients including those receiving high-flow oxygen	LMWH or UFH in prophylactic dose	LMWH or UFH in prophylactic dose	LMWH or UFH in prophylactic dose
Post- discharge	Against	Against	Rivaroxaban at a dose of 10 mg once daily in select patients

## Data Availability

Data supporting reported results can be found in National Tuberculosis and Lung Diseases Research Institute, 01-138 Warsaw, Poland.

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
