# Peer review of "Venous Thromboembolic Disease in COVID-19, Pathophysiology, Therapy and Prophylaxis"

_ijms, 2022, doi:10.3390/ijms231810372_

Round 1

Reviewer 1 Report

I read the review entitled “Venous thromboembolic disease in COVID-19, pathophysiology, therapy and prophylaxis” and my opinion rigarding this article is positive. Infact, the authors comprehensively describe all the known mechanisms of thrombogenesis in COVID-19 disease and exhaustively provide all the informations about the antithrombotic treatments. The artiche is written in a fluid way and it reads well. The paragraphs are well-structured and the images are interesting. It is clear that there is several literature about this topic. Anyway,  I do not feel like rejecting this manuscript on this basis. The authors made a good work of bibliography and a good work of writing and I think that this article may be useful to different specialists (hematologists, infectious diseases, internists). Therefore, I think that this manscript may be suitable for publication in its current version. I have only a recommendation to make regarding the section chosen by the authors entitled “Molecular Endocrinology and Metabolism” that I consider inappropriate. Threfore, I think that this manuscript should be submitted in a more appropriate section in IJMS.

Author Response

Dear Reviewer,

thank you very much for the positive review of our work. We are very pleased that you appreciated our efforts.

The article was written in response to an invitation to a special issue of "Frontiers in Thrombosis" and is thematically closely related to this topic.

"Frontiers in Thrombosis" is related to "Endocrinology and Molecular Metabolism", hence the connection.

Thank you very much again for your positive opinion

Yours faithfully,

The Authors

Reviewer 2 Report

The review by Dybowska et al gives a nice overview of literature and recommendations on VTE prophylaxis and treatment in COVID-19 patients.

I have the following suggestions for the authors:

1. You mention VTE risk in COVID-19 in the introduction, but I think you could add a little more detail about previous literature on VTE incidence/prevalence in COVID-19.

2. Tables summarizing the recommendations of existing guidelines on a) thromboprophylaxis and b) VTE treatment for different severities of COVID-19 (i.e. outpatients, inpatients, ICU patients) would be of great assistance to the reader

Author Response

Dear Reviewer,

we are very pleased that you appreciated our efforts.

Thank you very much for your comments. We made corrections as suggested.

Your suggestions for the authors:

  1. You mention VTE risk in COVID-19 in the introduction, but I think you could add a little more detail about previous literature on VTE incidence/prevalence in COVID-19.

---- Done, I am very thankful for the suggestion.

  1. Tables summarizing the recommendations of existing guidelines on a) thromboprophylaxis and b) VTE treatment for different severities of COVID-19 (i.e. outpatients, inpatients, ICU patients) would be of great assistance to the reader

---- a) I am very thankful for the suggestion. I have attached a table summarizing the recommendations for thromboprophylaxis in patients with COVID-19 without evidence of VTE.

---- b) As we wrote, VTE treatment is the some like for non- COVID-19 patients, but preferable are LMWH.

We hope you enjoy the article more readable now.

Thank you very much again for your positive opinion.

Yours faithfully,

The Authors